# Discovery of a Potential Human Serum Biomarker for Chronic Seafood Toxin Exposure Using an SPR Biosensor

**DOI:** 10.3390/toxins11050293

**Published:** 2019-05-23

**Authors:** Kathi A. Lefebvre, Betsy Jean Yakes, Elizabeth Frame, Preston Kendrick, Sara Shum, Nina Isoherranen, Bridget E. Ferriss, Alison Robertson, Alicia Hendrix, David J. Marcinek, Lynn Grattan

**Affiliations:** 1Environmental and Fisheries Sciences Division, Northwest Fisheries Science Center, National Marine Fisheries Service, National Oceanic and Atmospheric Administration, 2725 Montlake Blvd. East, Seattle, WA 98112, USA; kendrickps@gmail.com (P.K.); bridget.ferriss@noaa.gov (B.E.F.); 2U.S. Food and Drug Administration, Center for Food Safety and Applied Nutrition, College Park, MD 20740, USA; Betsy.Yakes@fda.hhs.gov; 3Aquatic Toxicology Unit, King County Environmental Laboratory, Seattle, WA 98119, USA; Elizabeth.Frame@kingcounty.gov; 4Department of Pharmaceutics, University of Washington, Seattle, WA 98195, USA; hms520@uw.edu (S.S.); ni2@uw.edu (N.I.); 5Department of Marine Sciences, University of South Alabama and the Dauphin Island Sea Lab, Dauphin Island, AL 36528, USA; arobertson@disl.org; 6Department of Environmental and Occupational Health Sciences, University of Washington, Seattle, WA 98105-6099, USA; aliciah1@uw.edu; 7Departments of Radiology and Bioengineering and Pathology, University of Washington Medical School, 850 Republican Street, Seattle, WA 98109, USA; dmarc@uw.edu; 8Neurology Department, University of Maryland School of Medicine, Baltimore, MD 21201, USA; LGrattan@som.umaryland.edu

**Keywords:** chronic exposure, environmental neurotoxin, serum biomarker, seafood toxin, algal toxin, marine biotoxin

## Abstract

Domoic acid (DA)-producing harmful algal blooms (HABs) have been present at unprecedented geographic extent and duration in recent years causing an increase in contamination of seafood by this common environmental neurotoxin. The toxin is responsible for the neurotoxic illness, amnesic shellfish poisoning (ASP), that is characterized by gastro-intestinal distress, seizures, memory loss, and death. Established seafood safety regulatory limits of 20 μg DA/g shellfish have been relatively successful at protecting human seafood consumers from short-term high-level exposures and episodes of acute ASP. Significant concerns, however, remain regarding the potential impact of repetitive low-level or chronic DA exposure for which there are no protections. Here, we report the novel discovery of a DA-specific antibody in the serum of chronically-exposed tribal shellfish harvesters from a region where DA is commonly detected at low levels in razor clams year-round. The toxin was also detected in tribal shellfish consumers’ urine samples confirming systemic DA exposure via consumption of legally-harvested razor clams. The presence of a DA-specific antibody in the serum of human shellfish consumers confirms long-term chronic DA exposure and may be useful as a diagnostic biomarker in a clinical setting. Adverse effects of chronic low-level DA exposure have been previously documented in laboratory animal studies and tribal razor clam consumers, underscoring the potential clinical impact of such a diagnostic biomarker for protecting human health. The discovery of this type of antibody response to chronic DA exposure has broader implications for other environmental neurotoxins of concern.

## 1. Introduction

Domoic acid (DA) is a neurotoxin that is naturally produced during harmful algal blooms (HABs) by toxigenic *Pseudo-nitzschia* species with global distributions [1]. During HABs the toxin is transferred through food webs via filter-feeding pelagic and benthic species of finfish, shellfish, and other invertebrates to marine mammals, seabirds, and humans causing severe neurotoxicity and mortality [2,3,4,5]. Domoic acid poisoning in humans has been termed amnesic shellfish poisoning (ASP) and is characterized by gastrointestinal distress, seizures, permanent anterograde memory loss, death, and a host of other permanent neurological symptoms [6]. The first documented ASP event occurred in 1987 when over 100 people became ill and 4 died after consuming DA-contaminated mussels [4,7]. Follow up analyses of toxin levels in meal remnants, as related to human symptomology and additional non-human primate laboratory exposure studies, were performed to set a general seafood safety regulatory limit of 20 μg DA/g seafood, established by Health Canada in the late 1980s and rapidly adopted by the US Food and Drug Administration [8]. The regulatory limit of 20 μg DA/g of edible shellfish tissue was designed to protect seafood consumers from DA doses that would cause visible symptoms after a single average shellfish meal. It does not consider the potential health effects of repetitive or chronic long-term exposure to lower toxin concentrations [9,10]. This raises concerns regarding the potential health risks of chronic DA exposure to putatively “safe” concentrations of DA that are not currently being considered in regulations.

Recent findings in laboratory models as well as seafood consumption studies in humans have called attention to the importance of considering chronic low-level DA exposure in the management of health risks. In controlled laboratory studies, subclinical neurologic effects have been reported after long term low-level exposure to DA at doses below those that elicit the obvious clinical signs of ASP. For example, repetitive low-level exposure caused learning deficits and hyperactivity in adult mice [11] as well as neurobehavioral changes in neonatal mice with low-level exposures in utero [12]. Studies in a nonhuman primate model point to additional effects of chronic low-level exposure. Female *Macaca fascicularis* monkeys were given daily oral doses of DA near the current allowable daily intake regulatory level for humans through pregnancy and gestation. Cognitive assessments in the offspring revealed significant consequences on emerging memory processes in neonates [13]. In addition, structural and chemical changes in brain morphology and the development of intentional tremors were observed over time in the adult females that were exposed during pregnancy [13,14]. The potential effects of chronic DA exposure may extend to humans as well. Consumption studies in coastal dwelling human seafood consumers in the Pacific Northwest, USA, have documented chronic DA exposure via razor clams (a shellfish species known to retain low levels of DA up to a year after a HAB in this region) in both recreational and tribal harvesters [10,15], as well as an association between chronic DA exposure via razor clam consumption and memory decline [16,17].

The urgency for assessing chronic low-level DA exposure and associated health risks is further underscored by the increase in geographic extent, duration, and severity of HABs. Warmer ocean conditions have been linked to more frequent, longer lasting, geographically larger, and more toxic *Pseudo-nitzschia* blooms, thereby increasing the risks of more frequent DA contamination of seafood resources [18,19,20]. Dietary exposures to the neurotoxin DA are inevitable for people consuming seafood harvested from regions where toxigenic *Pseudo-nitzschia* species occur, even with current regulatory limits enforced [10]. In order to effectively protect human health, both short-term recent exposure and long-term chronic exposure need to be considered when determining health risks and developing regulatory guidelines for safe seafood consumption. The subclinical effects of chronic low-level exposure to environmental neurotoxins are difficult to quantify in naturally exposed populations due to the multitude of confounding factors such as baseline health, alcohol and drug use, other contaminant exposures, age, and other general lifestyle choices. Unlike laboratory studies where exposure doses are controlled and defined, previous examinations of naturally exposed human populations have only been able to estimate DA exposure dose based on self-reported consumption rates and associated toxin levels in seafood reported by regulatory agencies. There is a critical need to develop tests for specific biomarkers that can verify chronic DA exposure in seafood consuming individuals.

In the present study, we collected two bodily fluids (blood and urine) in an effort to identify potential biomarkers of DA exposure in coastal dwelling Native Americans in Washington State who are at a particularly high risk of dietary DA exposure due to dependence on razor clams (RCs) as a food source. Razor clams in this region are known to contain low-levels of DA year-round [21]. In addition to bodily fluid samples, participants were given shellfish assessment surveys (SASs) to record recent RC consumption (within the last week) and long-term RC consumption (average monthly consumption over the last one to ten years). To test for a biomarker of chronic long-term DA exposure, human serum samples were evaluated for the presence of a DA-specific antibody. Additionally, human urine samples were tested for the presence of DA, which is rapidly eliminated via urine, to determine if DA is detectable after recent consumption and to confirm systemic DA exposure via consumption of legally-harvested shellfish.

## 2. Results

### 2.1. The DA Antibody Biomarker Was Detected in Some Chronic Shellfish Consumers via ELISA

Absorbance ratios indicative of DA-specific antibody presence were detected by enzyme-linked immunosorbent assay (ELISA) in three out of 42 serum samples collected in December 2015, four out of 40 samples collected in April 2016, and three out of 21 samples collected in May 2016. None of the serum samples collected in November 2012 (*n* = 29) were positive for antibody presence by ELISA. Absorbance ratios were calculated by dividing the RC consumer sample absorbance (X) by the sum of the mean non-RC consumer control serum absorbance and three times the standard deviation (X/0.153). Absorbance ratios greater than one in RC consumer serum samples indicated the presence of DA-specific antibodies. Control serum samples from non-RC consumers (*n* = 31) yielded a mean absorbance value of 0.078 ± 0.025 (SD). An additional blank (*n* = 6 over all assays), consisting of sample buffer only with no serum, yielded a mean absorbance value of 0.058 ± 0.002 (SD), suggesting that some non-specific binding occurs with control serum compared to buffer-only blanks. The small number of DA-specific antibody positive serum samples observed via ELISA revealed that a more sensitive method was needed for antibody detection. Consequently, a subset of samples was further analyzed via a surface plasmon resonance (SPR) biosensor.

### 2.2. The DA Antibody Biomarker Was Detected in a Majority of Chronic Shellfish Consumers via an SPR Biosensor

After the development of a more sensitive antibody detection method using a surface plasmon resonance (SPR) biosensor, a total of 61 serum samples from 22 individual RC consumers were analyzed via the SPR biosensor without the analyst’s knowledge of the participants’ consumption levels. All 22 individuals had serum drawn at a minimum of two and up to four possible timepoints (November 2012, December 2015, April 2016, and May 2016). After SPR analyses, long-term consumption was quantified for the 22 individuals. Twelve were categorized as high consumers (average consumption > 9 RCs per month over one to ten years), and ten were categorized as moderate consumers (average consumption of 3 to 9 RCs per month over one to ten years). Data from two of the high consumers were excluded from further analyses due to high nonspecific binding signals that prevented a determination of antibody presence or absence at multiple timepoints, and these results are not included in Table 1. Domoic acid-specific antibodies were detected in serum from at least one blood draw in 80% of the ten remaining high RC consumers and 40% of the moderate RC consumers tested (Table 1). In high consumers, 50% had the antibody present at all timepoints tested compared to 20% for moderate consumers (Table 1). Domoic acid-specific antibody presence was not detected in control samples (*n* = 17). Of the ten serum samples that tested positive by ELISA as described in Section 2.1., six of those also tested positive for antibody via SPR (Table 1). The other four ELISA-positive samples were not quantifiable via SPR due to high nonspecific binding (Table 1; participant ID 3 and another participant not included in Table 1 due to high non-specific binding at all timepoints). It is important to note that these reported percentages for antibody presence in chronic consumers are not representative of the population level prevalence due to limitations on the number of participants available for multiple blood draws, but these results do provide solid evidence for the development of the DA-specific antibody as a diagnostic biomarker.

The SPR biosensor was a better method than ELISA for detecting DA-specific antibodies due to the increased specificity, ability to use smaller sample volumes, and incorporation of multiple tests per sample. When the serum antibody binds to the DA on the SPR chip surface, so can nonspecific interactants. This can make it difficult to determine if binding is due to a specific surface-bound DA and antibody interaction. The use of a secondary antibody enabled both signal enhancement and signal interference reduction, as the anti-human secondary antibody only binds to the human antibody from serum and not to the nonspecific binders (e.g., non-immunoglobulin serum proteins). The use of multiple tests per sample with three requirements for a positive result for DA-specific antibodies improved confidence and detection sensitivity.

In order to determine if a sample contained DA-specific antibodies, three questions were evaluated: (1) is there something that binds to the DA chip, (2) is that binding specific or nonspecific, and (3) is the binding to DA versus the chip? Two assays were performed for each sample in order to fully answer these questions. The first assay was a direct evaluation of antibody binding from serum. In order to be categorized as having antibody, the sample must show a binding response higher than that of the controls (direct binding; answering question 1) and also have the secondary antibody binding response higher than the controls (answering question 2; Figure 1). Both of these were quantified via the change in refractive index upon binding that is greater than control refractive index changes. The second assay used a high concentration of DA (10,000 ng/mL) pre-mixed with the serum sample in an inhibition assay format to determine if the serum antibody showed solution specificity to DA. A decrease in binding of the DA-mixed versus the non-DA mixed samples indicated that the binding seen on the SPR sensor surface was for DA and not something else on the sensor surface (answering question 3). When all three requirements were met, the sample was determined to contain DA antibody (Figure 1).

### 2.3. Recent RC Consumption and Detection of DA in Urine Confirms Systemic Exposure

Domoic acid was detected in urine samples from participants reported to have recently consumed RCs, confirming that systemic DA exposure occurs with consumption of RCs containing DA below the regulatory threshold of 20 μg DA/g shellfish (Figure 2). The percentage of people who had consumed RCs during the recent consumption target period (last nine days for December 2015 participants, and last seven days for April and May 2016 participants) were 39%, 40%, and 44% in December 2015 (*n* = 123 total), April 2016 (*n* = 69 total,) and May 2016 (*n* = 18 total), respectively. The percentages of people reporting recent RC consumption and with DA positive urine samples were 21%, 50%, and 25% for December 2015, April 2016, and May 2016, respectively (Figure 2). All urine samples that tested positive for DA via ELISA and high performance liquid chromatography/mass spectrometry (HPLC-MS/MS) (*n* = 30) were significantly positively correlated (Figure 3). The lower limits of quantitation (LLOQ) for urine samples were 0.4 ng/mL for ELISA and 0.3 ng/mL for HPLC-MS/MS. Figure 4 depicts chromatograms of the three DA transitions in spiked blank human urine and in human urine from RC consumers testing positive for DA. A subset of 16 urine samples that tested positive for DA above 1 ng DA/mL by ELISA and should have been detectable by HPLC-MS/MS, were negative by HPLC-MS/MS, suggesting that false positives can occur via ELISA methods and that HPLC-MS/MS is required for validation.

Total recent DA exposure for each individual was calculated from the SAS based on the number of RCs consumed (converted to total number of grams using 45 g edible meat per clam [9]) and the average DA concentrations reported in RCs for corresponding time periods and tribal harvest beaches by the Washington Department of Health (WDOH) biotoxin monitoring program (23 ppm in December 2015, 6 ppm in April 2016, and 10 ppm in May 2016). Maximum DA exposures were 35, 11, and 6.3 mg of DA per person for target periods in December 2015, April 2016, and May 2016, respectively. Total milligrams of DA consumed per person were positively correlated with DA concentrations detected in corresponding urine samples in December 2015 and April 2016 (Figure 4). Domoic acid exposure and urine DA concentrations were not significantly correlated in the May samples due to a small sample size (*n* = 18) and a smaller range of DA exposures compared to the December (*n* = 123) and April (*n* = 69) timepoints (Figure 5).

## 3. Discussion

This study is the first to provide evidence for a DA-specific antibody response to chronic DA exposure in humans and, to our knowledge, is the first to detect and quantify the toxin in urine of naturally-exposed human seafood consumers. The detection of DA in urine and a DA-specific antibody in serum in multiple human shellfish consumers unequivocally confirms that Pacific Northwest coastal RC harvesters are systemically-exposed to DA via consumption of shellfish that contain toxin concentrations below the seafood safety regulatory limit and therefore deemed “safe” to consume. Moreover, these measures may serve as important biomarkers for diagnosing both recent and long-term chronic exposure, respectively (Figure 2; Table 1). Increased understanding of the exposure rates and the health impacts of both acute, high-level and chronic, low-level DA exposure are critical for effectively managing health risks to seafood consumers.

### 3.1. Detection of a DA-Specific Antibody in Serum Indicates Chronic Exposure

The detection of a DA-specific antibody in serum from multiple long-term RC consumers reveals a promising diagnostic tool for identifying chronically-exposed people. The first evidence for an immune response and DA-specific antibody production with chronic DA exposure was found during controlled laboratory studies by our research team using the zebrafish (*Danio rerio*) model system. Zebrafish were exposed once a week for multiple months to low doses of DA that were below doses that induce visible signs of neurotoxicity. Chronic DA exposure to these subclinical doses induced a significant immune response as indicated at the transcriptional level by whole-genome microarray profiling that was temporally linked to evidence of DA-specific antibody presence in serum [22]. Immune function genes were significantly upregulated after 18 weeks of exposure followed by evidence for the antibody in serum detected at the next sampling timepoint of 24 weeks of exposure, suggesting multiple weeks of low-level exposure were required for antibody production [22]. This discovery that long-term low-level DA exposure may lead to the induction of DA-specific antibody production in serum provided the impetus for exploring its use as a biomarker for chronic exposure in mammalian species using naturally-exposed California sea lions (*Zalophus californianus*). We found that 65% of sea lions known to have previous DA exposure tested positive for antibody presence while none of the non-exposed captive control sea lions tested positive, providing further validation of the potential biomarker [22]. These early studies in zebrafish and sea lions utilized ELISA techniques for antibody recognition and laid the groundwork for our investigations in human consumers. In the present study, we developed more sensitive and specific detection methods using an SPR biosensor that has provided compelling evidence for the antibody response in naturally exposed humans (Table 1).

The apparent transient nature of the DA-specific antibody presence in serum of chronic RC consumers is an attribute that makes it potentially more valuable as a clinical diagnostic biomarker (Table 1). All serum samples used for antibody detection were analyzed without knowledge of consumption status. We selected individuals for whom we had been able to collect serum at two or more timepoints in order to assess the consistency of antibody presence. After determination of antibody presence or absence in serum, long-term consumption data from the SASs were then used to determine chronic consumption status for each individual (Table 1). Fifty percent of high chronic consumers (those eating on average greater than 9 RCs per month year-round) and 20% of moderate RC consumers (those eating an average of 3 to 9 RCs per month year-round) tested positive for antibody presence at all timepoints. This is consistent with the assumption that greater exposure would be linked with increased prevalence of the antibody. In both high and moderate consumer categories, some individuals did not have detectable DA-specific antibody presence at any timepoint, while some were positive at some timepoints and negative at others (Table 1). These findings are consistent with the idea that antibody presence (and likely concentration) will vary in a predictable relationship with chronic exposure levels and duration. This is a critical element for establishing a biomarker of exposure. While we do not have the statistical power in this study to determine this relationship, our data provide strong evidence for the value of future studies with a larger population size.

### 3.2. Uses for a Biomarker of Chronic DA Exposure

The proposed DA-specific antibody biomarker would be valuable for identifying chronic exposure risks and, potentially, as a diagnostic indicator of underlying toxicological insult. Previous work in laboratory mouse models, using chronic low-level exposure paradigms, have documented novel effects of chronic exposure to DA at doses below those that induce visible outward signs of toxicity that are characteristic of ASP. These neurobehavioral effects of chronic low-level DA exposure included significant spatial learning and memory deficits as well as hyperactivity [11]. Unlike the permanent neurologic damage and hippocampal lesions documented with high-level seizure-inducing DA exposure [23,24,25], both hyperactivity and cognitive deficits observed with chronic low-level exposure were found to be reversible after a recovery period of no exposure and there were no gross morphologic or neuroinflammatory alterations in the hippocampal regions [11,26]. These reversible neurobehavioral effects were associated with a selective increase in vesicular glutamate transporter (VGluT1) levels within VGluT1-expressing boutons in the CA1 region of the hippocampus following repeated, low-level DA exposure in this model [26]. This could be responsible for creating a more excitatory environment in the brain through increased glutamate release from presynaptic excitatory boutons, thereby contributing to the observed hyperactivity and cognitive deficits observed [26]. If these patterns hold true for human seafood consumers, and if the DA-specific antibody is elevated in serum when cognitive deficits are present, but absent or reduced after recovery, then the biomarker would be a valuable diagnostic tool for chronic disease and subsequent recovery.

### 3.3. Detection of DA in Urine Confirms Systemic Exposure and Recent Consumption

The detection of DA in urine confirms that systemic DA exposure occurs via consumption of shellfish containing DA concentrations below current regulatory thresholds (20 μg DA/g) and may be useful for indicating recent exposure if sampled soon after consumption. The high variability of DA presence and level in RC consumer urine suggests that DA levels in urine are not accurate indicators of recent exposure over a seven-to-nine day period (Figure 2 and Figure 5). However, the fact that DA was detectable with consumption of legally-harvested seafood, suggests that detection of DA in urine may be useful as a shorter-term marker (i.e., within 24 h of consumption) for acute exposure concerns. Several factors confounded the relationship between DA concentrations in urine and DA consumption as quantified in the present study, such as toxin excretion rates, time after last meal, and rates of urination between exposure and sample collection. Domoic acid is rapidly excreted in the urine of mammals with a majority of the toxin eliminated within 24 h [27]. Rapid depuration rates along with the other listed factors account for the high variability in the presence and level of toxin detected in urine. The original goal was to obtain enough data to make comparisons of DA concentrations in urine and DA consumption within the last 24 h. However, the participant group taking the recent shellfish consumption surveys did not contain enough individuals with consumption within the last 24 h, thus requiring us to consolidate consumption over the week-long period. Even with a longer time period, DA was detectable in urine and correlated to consumption rates (Figure 5). We expect that urine sampled within 24 h of exposure would provide more consistent results, suggesting that DA in urine may be valuable as a diagnostic for acute exposure if urine is collected immediately after a patient feels ill from consuming shellfish. The use of urine DA tests would be a valuable component to assessing potential cases of DA neuroexcitotoxicity if collected within a short time window after suspected exposure. Indeed, domoic acid concentrations in urine, stomach contents, and feces have been used in assessments of marine mammal health for years and are part of a diagnostic protocol along with several other metrics for identifying acute DA poisoning in California sea lions [28,29].

## 4. Conclusions

In summary, our discovery of a DA-specific antibody in the serum of shellfish consumers is a breakthrough that could be used for the development of a diagnostic tool for assessing chronic DA exposure risks for which there are currently no protections. Our novel detection of DA in urine of naturally-exposed human shellfish consumers confirms that systemic DA exposure occurs via consumption of legally-harvested shellfish and represents a chronic exposure risk. The presence of a DA-specific antibody in serum provides an additional tool for studies to assess the relationship between chronic DA exposure and cognitive function in naturally exposed human populations. Future studies will also address whether the reduction or absence of the DA antibody is indicative of recovery following chronic DA exposure.

## 5. Materials and Methods

### 5.1. Quantification of Long-Term RC Consumption and Chronic Exposure

Shellfish assessment surveys (SAS) [30] to quantify long term razor clam (RC) consumption were administered to coastal dwelling Native Americans in Washington State as part of the “communities advancing the studies of tribal nations across their lifespan” (CoASTAL) study [31]. The CoASTAL participants completed yearly SASs, recording monthly RC consumption for as many as ten consecutive years. Average monthly RC consumption rates over one to ten years were then used to identify high consumers (those eating > nine RCs per month year-round) or moderate consumers (those eating three to nine RCs per month year-round).

### 5.2. Blood Collection for Biomarkers of Long-Term Chronic DA Exposure

Serum samples were collected from individuals at four sampling periods in conjunction with the CoASTAL study: in November 2012, December 2015, April 2016, and May 2016. Blood samples were collected by certified phlebotomists using BD Vacutainer Blood Collection kits into BD Vacutainer serum collection tubes. Whole blood was allowed to sit undisturbed at room temperature for 15–30 min to allow for clotting, followed by centrifugation at 4000× *g* for 15 min in a refrigerated centrifuge to remove the clot. Serum was immediately transferred and aliquoted into clean cryovials and frozen at −20 °C until further analyses via ELISA and/or SPR biosensor methods. An additional set of serum samples (*n* = 31) collected from moderate seafood consumers in Florida, where DA is not commonly observed, were opportunistically obtained and used as controls. Control blood samples were collected from volunteers with BD Vacutainer Safety-Lok blood collection kits into BD Vacutainer K2 EDTA spray coated tubes. Samples were mixed by inversion 8–10 times immediately after collection to ensure protein stabilization and anticoagulant action and then centrifuged at 2000× *g* for 15 min at 4 °C to separate serum and deplete platelets. Serum was aseptically removed and stored as aliquots at −20 °C. Before all analyses, serum samples were purified with a NAb Protein G Spin Kit per the vendor protocols (Thermo Fisher Scientific, Waltham, MA, USA) in order to remove extraneous proteins that may interfere with antibody binding and to enable an appropriate buffer for analysis. Cleaned-up serum was eluted in the standard amine-based elution buffer (pH 2.8) that was then neutralized with 10% (*v*/*v*) 1 M Tris HCl, pH 8.5 buffer.

### 5.3. Domoic Acid-Specific Antibody Presence via ELISA

Serum samples from RC consumers from Washington State collected in November 2012 (*n* = 29), December 2015 (*n* = 42), April 2016 (*n* = 40), and May 2016 (*n* = 21), as well as control serum samples (*n* = 31) collected from seafood consumers from Florida were analyzed for the presence of a DA-specific antibody via an enzyme-linked immunosorbent assay (ELISA) format. Detection specificity of DA-specific human IgG was accomplished using a DA conjugated 96-well plate obtained from the commercially available ASP ELISA kit for quantitative determination of domoic acid from Biosense Laboratories. Serum was applied to sample wells and DA-specific antibodies were allowed to bind to the DA conjugated to the plate. After thorough washing, HRP labeled anti-human IgG (goat anti human IgG H + L (HRP) from abcam, ab97161) diluted 1:5000 in 1% ovalbumin in PBS-T was allowed to incubate for one hour at room temperature, then TMB substrate was added, and absorbance was detected with a BioTek Epoch Plate reader. Absorbance ratios greater than one were used to determine antibody presence and were calculated by dividing RC consumer serum sample absorbance by the mean control absorbance plus three times the standard deviation from 31 control serum samples.

### 5.4. Domoic Acid-Specific Antibody Presence via an SPR Biosensor

A subset of control serum samples (*n* = 17) and serum samples from RC consumers (*n* = 22) in which samples were taken at more than one timepoint for each individual were analyzed for the presence of a DA-specific antibody via a surface plasmon resonance (SPR) biosensor (T200, GE Healthcare, Pittsburgh, PA, USA). Samples were run in random order during three consecutive days using a previously developed SPR biosensor surface [32] and a modified assay for detecting DA antibody in the complex serum matrix. The sensor surface had three DA-conjugated channels and one reference channel. Each sample was injected over all four flow cells, secondary antibody was then introduced, and then regeneration solution pulled both off the complex leaving the DA surface ready for the next sample.

Two vials of cleaned-up serum were diluted 1:5 in HBS-EP+ (GE Healthcare, pH 7.4) running buffer. Two aliquots of samples were used: the first aliquot vial was mixed with 10% (*v*/*v*) with 0 ng DA/mL, while the second was mixed with 10,000 ng DA/mL (Sigma-Aldrich, St. Louis, MO, USA; DA diluted in HBS-EP+) just prior to analysis. This allowed for two analyses of each sample with the first being a direct binding assay and the second being an inhibition assay evaluation. As shown in Figure 6, a sample cycle consisted of a binding time of 120 s (flow rate of 25 µL/min), followed by a 90 s binding of secondary antibody (30 µg/mL goat anti-human IgG H&L (abcam, ab97161), flow rate of 10 µL/min), and then regeneration for 60 s with 50 mM HCl, 0.5% SDS (flow rate of 25 µL/min). The secondary antibody was employed to increase the signal intensity, while simultaneously lowering background interference from non-specific binding. Positive control samples of mouse anti-DA (1:800 dilution in HBS-EP+; information on how this antibody was produced and evaluated is found in reference 32) with 30 µg/mL secondary, rabbit anti-mouse (GE Healthcare, mouse antibody capture kit, BR-1008-38), and negative control samples (buffer and non-exposed human serum) were interspersed with the RC consumer and control samples to ensure assay functionality and chip stability.

To evaluate the data, control sample averages and standard deviations were obtained to create cutoff values for direct binding (response at end of the serum injection minus the starting baseline), secondary antibody binding (response at the end of the secondary antibody injection minus the starting baseline), and amount of inhibition (response difference between binding for the 0 ng/mL versus the 10,000 ng/mL DA in serum samples; Figure 1). Controls run on Days 1 and 2 (*n* = 8) were used to determine an average cutoff value for the initial binding, secondary antibody, and inhibition responses. As the chip showed minor degradation in binding in the positive control, mouse-DA antibody on Day 3, separate control samples (*n* = 11) analyzed on Day 3 were used to create the corresponding cutoff values for data generated on Day 3.

### 5.5. Quantification of Recent RC Consumption and DA Exposure

Shellfish assessment surveys, designed to quantify recent RC consumption, were also completed by CoASTAL participants at three sampling dates. In order to quantify recent DA exposure, the number of RCs consumed, as well as the source beaches for harvesting, were recorded for target periods of the last nine days for December 2015 surveys and the last seven days for April and May 2016 surveys. Recent DA exposure was calculated for each individual using the total number of grams of RCs consumed in the target period multiplied by the DA levels quantified in RCs from the source harvest beaches, as reported by the Washington State Department of Health (WDOH) Biotoxin Monitoring Program.

### 5.6. Quantification of DA in Urine as an Indicator of Systemic Exposure and Recent Consumption

Urine samples were collected in sterile 100 mL urine collection cups from individuals on the same day that they completed the recent RC consumption surveys (*n* = 123 in December 2015, *n* = 69 in April 2016, and *n* = 18 in May 2016). Immediately following collection, urine samples were cooled on ice and frozen with dry ice until placed in laboratory freezers at −20 °C. Domoic acid was quantified in urine samples using a commercially available enzyme-linked immunosorbent assay (ELISA) for DA from Biosense as per kit instructions [33] and validated via high performance liquid chromatography tandem mass spectrometry (HPLC-MS/MS). For ELISA methods, DA was extracted from urine via standard procedures using a 1:4 ratio of sample to 50% MeOH extraction solvent [34]. Final extracts were further diluted 10-fold in dilution buffer before quantification. For HPLC-MS/MS analyses, the samples were analyzed using a recently developed and validated HPLC-MS/MS method [35]. In brief, urine samples were extracted with 100% methanol at 1:1 *v*/*v* ratio. Samples were vortexed for 15 s and subsequently centrifuged at 16,100× *g* for 15 min. Supernatant was collected for analysis. Standard curves were prepared in naïve human urine with spiked domoic acid concentration ranging between 0.3 and 40 ng/mL. Samples were analyzed on Shimadzu UFLC XR DGU-20A5 (Shimadzu Scientific Instruments, Pleasanton, CA, USA) equipped with Synergi Hydro-RP 100 Å LC column (2.5 µm, 50 mm × 2 mm; Phenomenex) with a guard cartridge (2 × 2.1 mm, sub 2 µm; Phenomenex, Torrance, CA, USA) connected in line with AB Sciex 6500 qTrap Q-LIT mass spectrometer (AB Sciex, Concord, Ontario, Canada). The HPLC method uses a 9 min gradient running at 0.5 mL/min. The gradient initiates at 95% (A) water with 0.1 formic acid and 5% (B) 95:5 (*v*/*v*) acetonitrile:water with 0.1% formic acid for a minute, gradually increases to 100% B over the next 3 min, continues at 100% B for 30 s before returning to 5% B over the next 30 s, and runs at initial condition for another 4 min. Domoic acid was ionized using electrospray ionization (ESI) operating in positive ion mode and was monitored using multiple reaction monitoring (MRM) for *m*/*z* transition 312.1 > 266.1, *m*/*z* 321.1 > 248.1, and 312.1 > 220.1 (Figure 4). The urine calibration standards and quality control samples were prepared by spiking blank urine with the authentic certified reference standards as described previously.

## Figures and Tables

**Figure 1 toxins-11-00293-f001:**
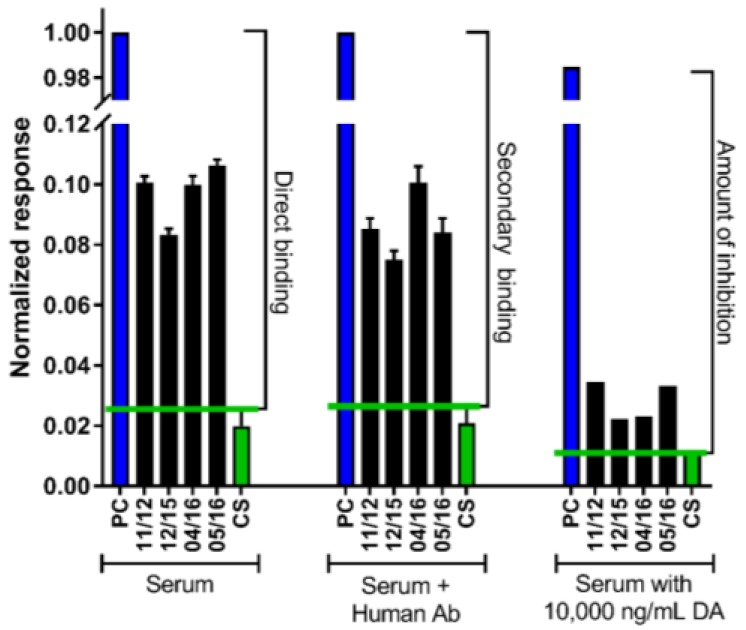
Example graph showing positive detection of a domoic acid (DA)-specific antibody (Ab) by surface plasmon resonance (SPR) in serum collected at four timepoints from a “high” chronic razor clam (RC) consumer (High = greater than nine RCs per month year-round; ID 1 in Table 1). The blue bars represent the mouse DA Ab positive control samples (PC). Black bars represent human seafood consumer serum at four collection dates (November 2012, December 2015, April 2016, and May 2016). The green bars represent controls (CS; human serum from seafood consumers from a region without DA blooms). Three analyses are shown: (1) direct binding, (2) secondary Ab binding, and (3) amount of inhibition. Horizontal green lines denote cutoff values based on control serum for direct binding (response at the end of the serum injection minus the starting baseline), secondary antibody binding (response at the end of the secondary antibody injection minus the starting baseline), and amount of inhibition (response difference between binding for the 0 ng/mL versus the addition of 10,000 ng/mL DA in serum samples). Error bars represent the standard deviation from triplicate measurement, and the data are normalized to the mouse DA-specific Ab control responses.

**Figure 2 toxins-11-00293-f002:**
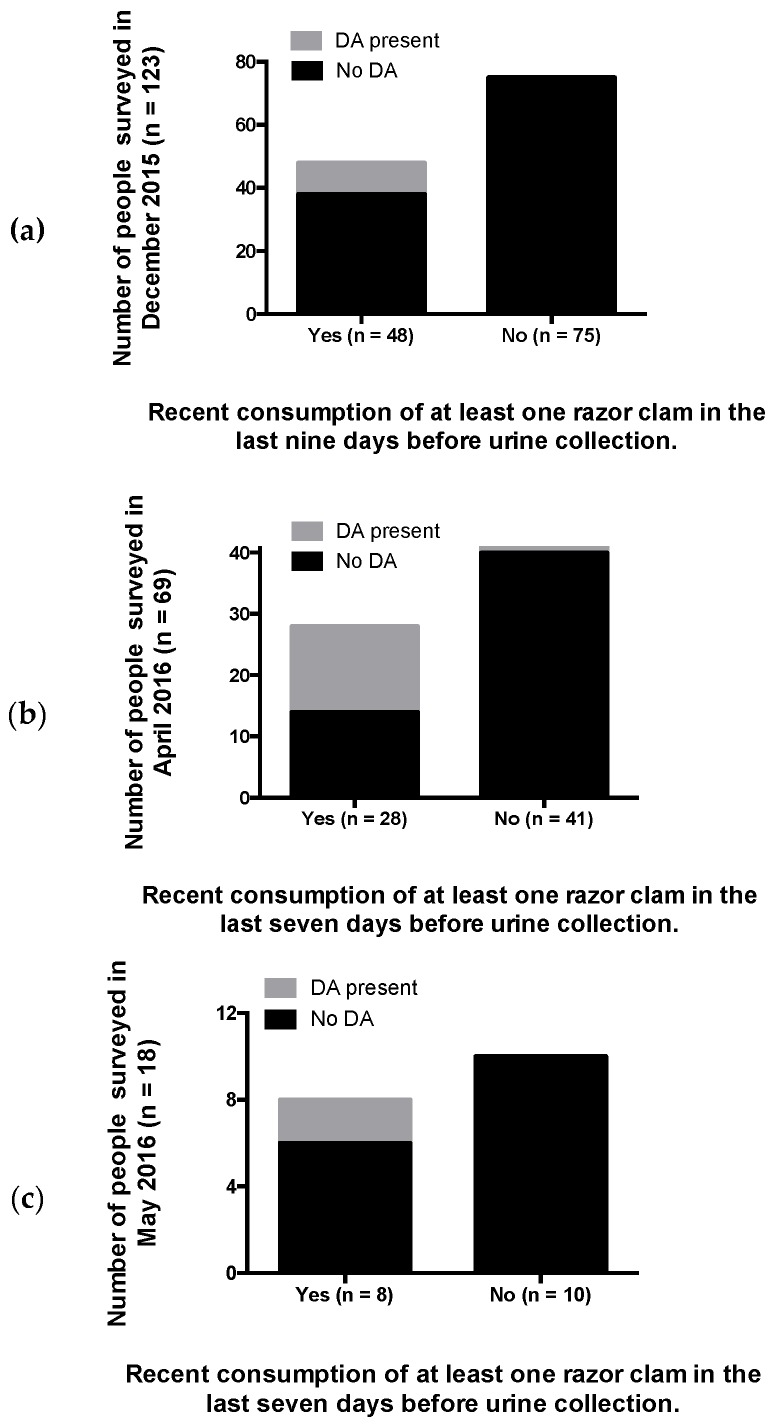
Bar graphs showing the number of people that consumed clams during the seven to nine days prior to urine sampling (Yes category) and the number of domoic acid (DA)-positive urine samples (gray portion in bar) collected in (**a**) December 2015, (**b**) April 2016, and (**c**) May 2016.

**Figure 3 toxins-11-00293-f003:**
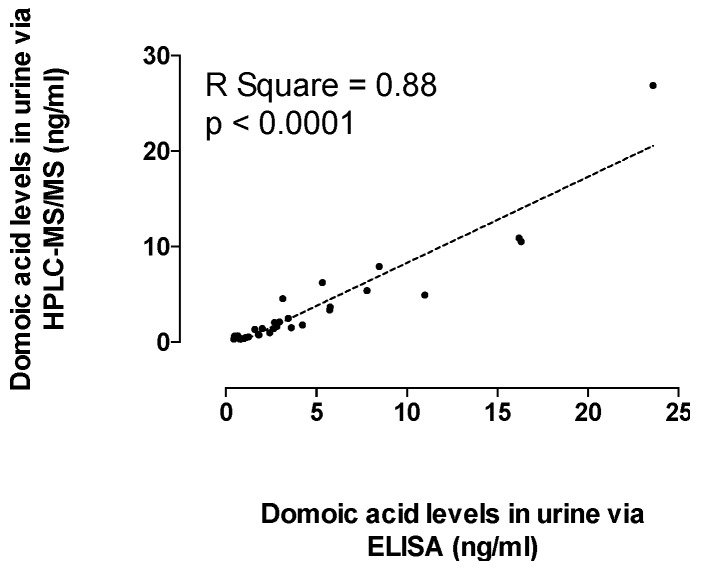
Comparison of domoic acid (DA) concentrations in urine samples (*n* = 30) quantified via Biosense DA enzyme-linked immunosorbent assays (ELISA) and high performance liquid chromatography tandem mass spectrometry (HPLC-MS/MS).

**Figure 4 toxins-11-00293-f004:**
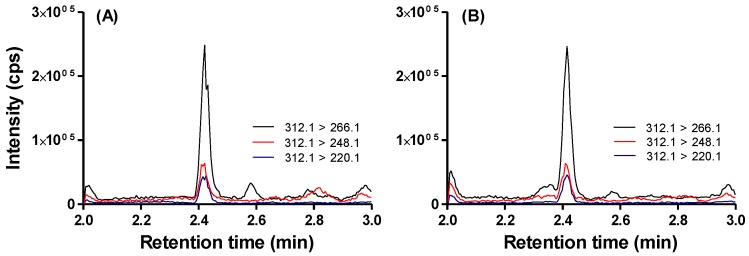
High performance liquid chromatography tandem mass spectrometry (HPLC-MS/MS) chromatograms of three domoic acid (DA) transitions (*m*/*z* 312.1 > 266.1, 248.1, 220.1) in (**A**) spiked urine with 19.9 ng DA/mL, and (**B**) human urine sample measured with 20.2 ng DA/mL.

**Figure 5 toxins-11-00293-f005:**
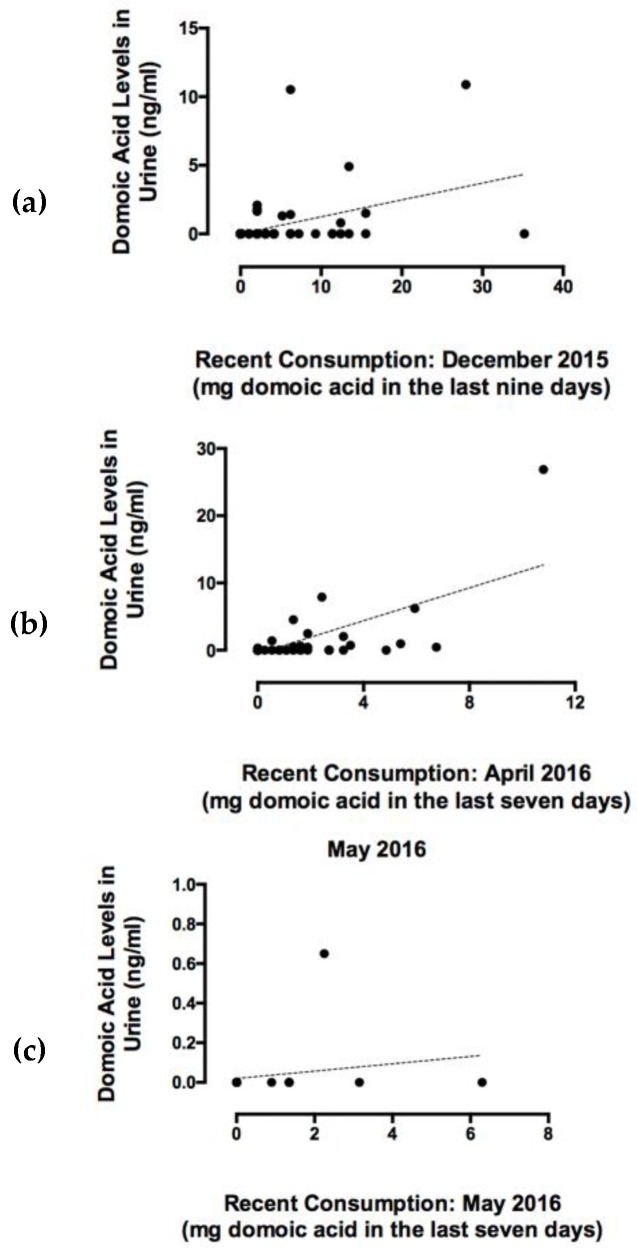
Comparison of total domoic acid exposure in the previous nine days in people surveyed in (**a**) December 2015, or previous seven days in people surveyed in (**b**) April and (**c**) May 2016, and toxin levels quantified via HPLC-MS/MS in corresponding urine samples collected on the same day that participants completed the recent consumption surveys.

**Figure 6 toxins-11-00293-f006:**
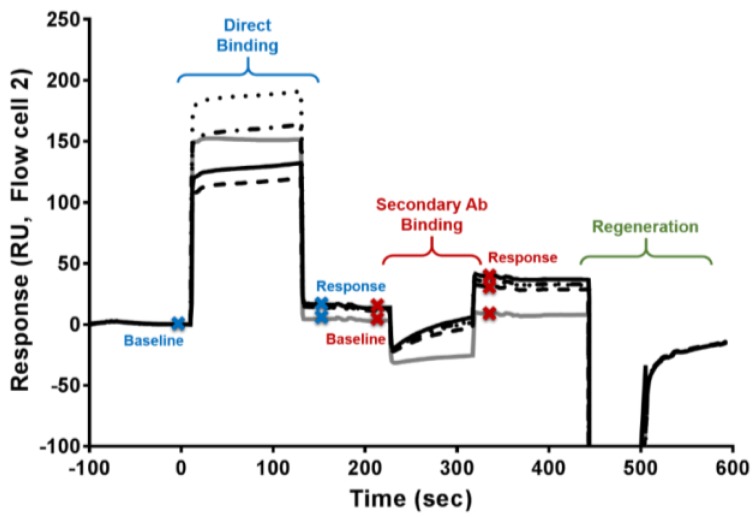
Overlay plot of sensorgrams showing the analysis cycle: direct binding, secondary antibody binding, and regeneration. Respective baselines and responses (blue = direct binding and red = secondary antibody binding) used to calculate the SPR signal are marked with “X” on the curves. The black lines (dotted, dash-dot, solid, and dashed) are representative sensorgrams from the four serum samples collected from a “high” RC consumer (ID 1 in Table 1 and Figure 1): November 2012 sample (dotted curve); December 2015 sample (dashed curve), April 2016 sample (solid curve), and May 2016 sample (dashed-dotted curve). The grey curve illustrates a representative sensorgram for a control serum sample (CS; human serum from a seafood consumer from a region without DA blooms; same sample as displayed in Figure 1). A simple visual evaluation illustrates that the “high” RC consumer, at all timespoints, had more SPR signal at both the direct binding response (blue Xs) and the secondary antibody binding response (red Xs) steps than the CS sample. These values are quantified and graphed in Figure 1.

**Table 1 toxins-11-00293-t001:** Presence of a domoic acid (DA)-specific antibody via a surface plasmon resonance (SPR) biosensor in multiple serum samples from 20 razor clam (RC) consumers. Consumers were classified as high or moderate consumers based on average monthly consumption rates recorded in yearly shellfish assessment surveys (SASs) over one to ten years. High = greater than nine RCs per month year-round, and moderate = three to nine RCs per month year-round.

Sample ID	Average Monthly Consumption	Antibody Presence (Nov 2012)	Antibody Presence (Dec 2015)	Antibody Presence (April 2016)	Antibody Presence (May 2016)	Years of SASs
1	High	**Yes**	**Yes**	**Yes**	**Yes**	9
2	High	**Yes**	**Yes**	**Yes**	*ns*	10
3	High	**Yes**	No	**+	**+	6
4	High	**Yes**	No	*ns*	*ns*	10
5	High	*ns*	**Yes**	**Yes**	**Yes**	2
6	High	*ns*	**Yes**+	**Yes**+	**Yes**+	1
7	High	*ns*	**Yes**+	**Yes**	*ns*	3
8	High	*ns*	No	**Yes**	No	2
9	High	*ns*	No	No	No	9
10	High	*ns*	No	No	*ns*	1
11	Moderate	**Yes**	**Yes**	*ns*	**Yes**+	8
12	Moderate	**Yes**	**Yes**	**Yes**+	*ns*	8
13	Moderate	**Yes**	**Yes**	No	*ns*	10
14	Moderate	**Yes**	No	*ns*	*ns*	6
15	Moderate	No	No	No	*ns*	7
16	Moderate	*ns*	No	No	No	1
17	Moderate	*ns*	No	No	No	1
18	Moderate	*ns*	No	*ns*	No	9
19	Moderate	*ns*	No	No	*ns*	2
20	Moderate	*ns*	No	No	*ns*	10

** = unable to determine due to high nonspecific binding; **YES** = tested positive for DA-specific antibody via an SPR biosensor; + = also tested positive for DA-specific antibody via ELISA; No = DA-specific antibody was not detected; *ns* = no sample available.

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
