# Peer review of "Discovery of a Potential Human Serum Biomarker for Chronic Seafood Toxin Exposure Using an SPR Biosensor"

_toxins, 2019, doi:10.3390/toxins11050293_

Round 1
Reviewer 1 Report
Line 341: --- 4,000 g; Line 349: --- 2,000 x g; Line 418: --- 16,100 g:
The discription in "5.3. Domoic acid specific antibody presence via ELISA (Line 327 ~ " seems to be insufficient. Details of the 96 well plate (company), how to conjugate DA on the plate (if already reported, please refer it), the origin of DA standard, HRP-labelled anti-Human IgG (company, dilution ready for use, etc), incubation time (min) / temperature etc, should be discribed here.
Author Response
Thank you for your review.
We have removed the x from Line 349 as requested.
We have also added the information you requested to make this a more complete methods section for the ELISA.
Detection specificity of DA-specific human IgG was accomplished using a DA conjugated 96-well plate obtained from the commercially available ASP ELISA kit for quantitative determination of domoic acid from Biosense Laboratories.Serum was applied to sample wells, and DA-specific antibodies were allowed to bind to the DA conjugated to the plate. After thorough washing, HRP labeled anti-human IgG (goat anti human IgG H + L (HRP) from abcam) diluted 1:5,000 in 1% ovalbumin in PBS-T was allowed to incubate for one hour at room temperature, thenTMB substrate was added, and absorbance was detected with a BioTekÒEpoch Plate reader.
Reviewer 2 Report
The manuscript entitled “Discovery of a Potential Human Serum Biomarker for Chronic Seafood Toxin Exposure Using an SPR Biosensor” reports for the first time an antibody against domoic acid that appears in chronic consumers of potentially contaminated seafood under the regulation limits. This is a highly relevant finding in terms of the development of new diagnostic tools to evaluate exposure and risk of domoic acid chronic sub-acute consumption. Additionally, this study reports for the first time the presence of domoic acid in human urine samples after seafood consumption.
According to this reviewer, only minor issues should be addressed before considering the manuscript for publication in Toxins.
Minor points:
- “An” in the title should be “a”?
- In table 1: + = also tested positive for DA specific antibody via ELISA. Were any of the samples positive for ELISA and negative for SPR? If not, this should be specifically stated.
- Materials and methods: which is the antibody used as positive control in ELISA for the normalization of the signal?
- Despite of being based on “Talanta, 2016. 156-157: p. 55-63” and more considering that the method described for SPR sensor is different (enhancement), representative sensograms should be presented for the different sample where all the steps are covered (binding, stationary and regeneration phases) for direct and competitive detection.
Author Response
Thank you for your review and helpful comments.
We have addressed the comments as follows:
1) The title now says "a" potential biomarker... instead of "an".
2) We added text in section 2.2. describing the ELISA positive samples (n=10) and how they tested via SPR.
Domoic acid-specific antibodies were detected in serum from at least one blood draw in 80% of the ten remaining high RC consumers and 40% of moderate RC consumers tested (Table 1). In high consumers, 50% had antibody present at all time points tested compared to 20% for moderate consumers (Table 1). Domoic acid specific antibody presence was not detected in control samples (n = 17). Of the ten serum samples that tested positive by ELISA as described in section 2.1., six of those also tested positive for antibody via SPR (Table 1). The other four ELISA-positive samples were not quantifiable via SPR due to high nonspecific binding (Table 1; participant #3 and another participant not included in Table 1 due to high non-specific binding at all time points).
3) We added more information regarding the positive control antibody:
Two vials of cleaned-up serum were diluted 1:5 in HBS-EP+ (GE Healthcare,pH 7.4) running buffer. Two aliquots of samples were used: the first aliquot vial was mixed with 10% (v/v) with 0 ng DA/mL, while the second was mixed with 10,000 ng DA/mL (Sigma DA diluted in HBS-EP+)just prior to analysis. This allows for two analyses of each sample with the first being a direct binding assay and the second being an inhibition assay evaluation. As shown in Figure 6,a sample cycle consisted of a binding time of 120 sec (flow rate of 25 µL/min), followed by 90 sec binding of secondary antibody (30 µg/mL goat anti-human IgG H&L (abcam, ab97161), flow rate of 10 µL/min), and then regeneration for 60 sec with 50 mM HCl, 0.5% SDS (flow rate of 25 µL/min). The secondary antibody was employed to increase the signal intensity, while simultaneously lowering background interference from non-specific binding. Positive control samples of mouse anti-DA (1:800 dilution in HBS-EP+; information on how this antibody was produced and evaluated is found in reference 32)with 30 µg/mL secondary, rabbit anti-mouse (GE Healthcare, mouse antibody capture kit, BR-1008-38)and negative control samples (buffer and non-exposed human serum) were interspersed with the RC consumer and control samples to ensure assay functionality and chip stability.
4) We added a figure of example sensorgrams as requested. See figure 6 and figure legend below:
Figure 6. Overlay plot of sensorgrams showing the analysis cycle: direct binding, secondary antibody binding, and regeneration. Respective baselines and responses (blue = direct binding and red = secondary antibody binding) used to calculate the SPR signal are marked with “X” on the curves. The black lines (dotted, dash-dot, solid, and dashed) are representative sensorgrams from the four serum samples collected from a “high” RC consumer (ID #1 in Table 1 and Figure 1): November 2012 sample (dotted curve); December 2015 sample (dashed curve); April 2016 sample (solid curve); and May 2016 sample (dotted-dashed curve). The grey curve illustrates a representative sensorgram for a control serum sample (CS; human serum from a seafood consumer from a region without DA blooms; same sample as displayed in Figure 1). A simple visual evaluation illustrates that the “high” RC consumer, at all times points, had more SPR signal at both the direct binding response (blue x’s) and the secondary antibody binding response (red x’s) steps than the CS sample. These values are quantified and graphed in Figure 1.